# Dual-Mode Conical Horn Antenna with 2-D Azimuthal Monopulse Pattern for Millimeter-Wave Applications [note 1]

**DOI:** 10.3390/s23198157

**Published:** 2023-09-28

**Authors:** Asrin Piroutiniya, Mohamad Hosein Rasekhmanesh, José Luis Masa-Campos, José Luis Calero-Rodríguez, Jorge A. Ruiz-Cruz

**Affiliations:** Group of RadioFrequency: Circuits, Antennas and Systems (RFCAS), Escuela Politécnica Superior, Universidad Autónoma de Madrid, 28049 Madrid, Spain; asrin.piroutiniya@estudiante.uam.es (A.P.); mohamad.rasekhmanesh@estudiante.uam.es (M.H.R.); joseluis.masa@uam.es (J.L.M.-C.); jose.calero@estudiante.uam.es (J.L.C.-R.)

**Keywords:** monopulse, dual-mode converter, mode converters, conical horn antenna, K-band, 5G, millimeter-wave applications

## Abstract

In this paper, a novel concept of a three-dimensional full metal system including a Dual-Mode Converter (DMC) network integrated with a high-gain Conical Horn Antenna (CHA) is presented. This system is designed for 5G millimeter wave applications requiring monopulse operation at K-band (37.5–39 GHz). The DMC integrates two mode converters. They excite either the $TE_{11}{}^{cir}$ or the $TE_{01}{}^{cir}$ modes of the circular waveguide of the CHA. The input of the mode converters is the $TE_{10}{}^{rec}$ mode of two independent WR-28 standard rectangular waveguide ports. By integrating the DMC with the CHA, the whole system, called a Dual-Mode Conical Horn Antenna (DM-CHA), is formed, radiating the sum (Σ) and difference (Δ) patterns associated to the monopulse operation. To adequately prevent the propagation of higher order modes and mode mutual coupling, this integration procedure is carefully designed and fabricated. To prove the performance of the design, the DMC network was fabricated using subtractive manufacturing by Computer Numerical Control (CNC) technology. The CHA was fabricated using additive manufacturing by Direct Metal Laser Sintering (DLMS) technology. Finally, the simulation and measurement results were exhaustively compared, including return loss, isolation, radiation pattern, and gain of the full DM-CHA structure. It is noteworthy that this system provided up to ±11° per beam in the angular of arrival detection to support the high data rate operation for 5G satellite communications in the millimeter-wave band.

## 1. Introduction

Monopulse antennas are commonly employed in radar applications for target identification and tracking [1,2]. Recently, monopulse antennas have also been suggested for 5G and millimeter wave applications in the millimeter-wave band to enhance the identification of user signals [3,4].

Many antenna structures have been proposed for monopulse systems during the past decades [5,6,7,8,9,10]. They include a radiating part and a feeding part, the latter of which is a specific feeder circuit in charge of properly exciting in the radiating part the following independent radiation diagrams: sum pattern Σ (for receiving and transmitting data) and the difference pattern Δ (just for receiving data). By amplitude comparison of both patterns, the user direction of arrival can be calculated with very high precision [11]. Depending on the topology and technology, they can be classified as planar two-dimensional (2D) or three-dimensional (3D) structures, as is the one proposed in this work. In the context of planar 2D structures, microstrip circuitry in the form of arrays are widely employed for a number of applications due to their small weight, low cost, and ease of fabrication [12,13,14,15,16,17,18,19]. In these cases, the radiating part is based on array topologies divided into at least two or more sections connected to the feeding part, which is generally composed of a two-input port network with couplers and phase shifters to generate an in-phase feeding of the array for the sum pattern while an opposite-phase is generated for the difference pattern. However, planar microstrip structures present high losses, which make them unsuitable especially for the millimeter-wave range.

Substrate integrated waveguide technology (SIW) is used to minimize the loss problems associated with microstrip array antennas while the compactness of a 2D structure is maintained. The same feeding network topology based on coupler circuits is also used in SIW monopulse designs from the Ka to W bands [20,21,22]. Nevertheless, as SIW circuits are based on waveguide topologies, this kind of monopulse feeding part takes up considerable space compared to the radiating part, which is mainly formed by slotted waveguide arrays printed on the same substrate layer. In order to avoid this space problem and reduce the complexity, monopulse feeding networks based on the excitation of two [23,24] or more [25] high-order modes on the printed waveguide have been proposed, which on occasion can be implemented even in multilayer disposition [26]. On the other hand, these multimode feeding networks present a narrower bandwidth response.

In satellite communications, high-gain antennas are necessary due to the long link distance. For this reason, the required bigger size of 2D planar array configurations leads to an efficiency deterioration, especially in the millimeter band, due to the associated losses of the printed monopulse feeding network, even with the use of SIW technology. In recent years, 2.5D intricated multilayer planar structures implemented in full metallic gap waveguide technology have been proposed in the millimeter band (up to W band [27,28]) as a solution to minimize feeding network losses in high-gain array monopulse systems, mainly applied to terrestrial terminals but with a significant cost increase.

On-board satellite antenna systems, in which a broader frequency band is required, are mostly implemented with 3D systems, mainly composed of a single directive horn antenna [12,13] in the radiating part (adding a reflector structure if higher gain is required [29,30,31]) and metallic waveguide devices in the feeding part. In these single antenna systems, the monopulse performance can only be achieved with dual (or multi-) mode networks by means of two main topologies: multimode segmented (generally four) waveguides with different phase sector combinations to generate the sum Σ and the difference Δ patterns [32,33,34,35], or non-segmented waveguides with several (at least two) mode excitation circuits (mode converters) to generate the monopulse pattern [36,37,38]. Both feeding topologies commonly are bulky structures, especially if other extra capabilities are implemented (such as cross-polarization cancelation [39] or filtering properties [40]) by multimode combination networks. For simplicity, the commonly selected horn antennas for the radiating part are pyramidal [32,33,34] or conical [36,37,38]. Nevertheless, coaxial horn antennas are also an interesting choice for multiband operation [41], as well as corrugated designs for radiation pattern angular symmetry [42].

In this paper, an on-board satellite linearly polarized Conical Horn Antenna (CHA) is presented for millimeter band 5G communications. In addition, identification of the user direction of signal arrival is provided by means of a monopulse feeding network. As cellular communications are massive user systems, the cost is an important design factor, which is directly related to the system complexity. For this reason, a compact and simple non-segmented dual mode topology has been implemented for the monopulse feeding part. Thus, a circular waveguide will be its out port connecting with the CHA and involving ideally only two modes: the $TE_{11}{}^{cir}$ and the $TE_{01}{}^{cir}$. Nominal rectangular waveguides will be used as antenna system input ports. Therefore, rectangular-to-circular mode converters will be required.

In this work, the design aims to produce the Σ and Δ patterns in the following way. The sum Σ pattern will be generated when the fundamental mode of the circular waveguide $(TE_{11}{}^{cir}$), working in the vertical polarization, excites the horn input. The difference Δ pattern will be generated when the $TE_{01}{}^{cir}$ excites the horn input. Please note that the $TE_{01}{}^{cir}$ is a high-order mode of the circular waveguide, and the excitation levels of other waveguide modes with a cutoff frequency between the $TE_{11}{}^{cir}$ and the $TE_{01}{}^{cir}$ (and those close above) will have to be very low to not degrade the system performance. This will be achieved in the presented design using a proper control of the symmetries of the structure and the feeder networks.

The presented design is a suitable candidate for the future generation of 5G millimeter-wave applications requiring high data rates. Future 5G high throughput satellites can use the suggested antenna as a component of an on-board multi-spot array [43], in which the monopulse tracking method [4] can be used to estimate the signal angle at each satellite spot. The setup creates a beam with a high gain and low side lobes as well as a very deep null in the boresight.

The paper is organized as follows. The design process is divided into two sections (Section 2 and Section 3). In Section 2, the mode converters ($TE_{10}{}^{rec}$ to $TE_{11}{}^{cir}$ and $TE_{10}{}^{rec}$ to $TE_{01}{}^{cir}$) are designed individually. In Section 3, the integration of the two mode converters to provide the DMC is described along with the connection to the CHA. The fabrication and experimental results are described in Section 4, validating the presented approach.

## 2. Design of the Mode Converters for the Sum and Difference Patterns

The monopulse system will have two dedicated ports in rectangular waveguide mode, associated to the Σ and Δ patterns, respectively. They operate in their fundamental $TE_{10}{}^{rec}$ rectangular waveguide modes. Thus, conversion between modes of the rectangular and circular waveguides is required. For the conversion between the $TE_{10}{}^{rec}$ and the $TE_{11}{}^{cir}$ modes in the sum Σ pattern, a novel geometry will be introduced. This conversion only involves fundamental modes. It is therefore simpler than that required for the difference Δ pattern. In this last case, the conversion to the $TE_{01}{}^{cir}$ mode is required, avoiding as much as possible the generation of other non-desired modes in the circular waveguide. There are several methods to design mode converters involving the $TE_{01}{}^{cir}$ mode [44,45,46,47,48,49,50], which are used in different applications. The side wall coupling used for instance in [46,48] and the inline coupling in [49,50,51] are two suitable candidates. The sidewall coupling approach was utilized in this work since it leads to shorter and more compact designs, while allowing a dedicated control of the spurious excitation of non-desired higher order modes.

Both mode converters between the $TE_{10}{}^{rec}$ and the $TE_{11}{}^{cir}$,$\;TE_{01}{}^{cir}$ modes have to be integrated within the same horn feeder, making up a novel dual-mode converter (DMC) feeding network. The DMC will excite a dual-mode conical horn antenna with a 2-D azimuthal monopulse pattern (vertical polarization) in the frequency band of 37.5–39 GHz. Initial work was carried out by [52]. Here, the full design methodology is described, detailing the fabrication process and the experimental results, which will be compared with the theoretical ones. The CHA is fabricated using additive manufacturing by Direct Metal Laser Sintering (DMLS). The DMC is fabricated using subtractive manufacturing by a standard Computer Numerical Control (CNC) process.

In this section, we present the design process of the two mode converters found in the system. One is associated to the sum pattern, while the other is associated to the difference pattern. In this Section 2, the designs are carried out independently. Section 3 later on explains how they are integrated to provide the DMC that is connected to the Conical Horn Antenna (CHA).

### 2.1. Mode Converter from ${TE}_{10}{}^{rec}$ Mode in the Rectangular Waveguide to the $TE_{11}{}^{cir}$ Mode in the Circular Waveguide

The proposed $TE_{10}{}^{rec}–TE_{11}{}^{cir}$ mode converter is shown in Figure 1. Only the inner air channel is represented, without the metallic enclosure. Since the converter only involves fundamental modes at the input and output ports, a gradual transition is used [53,54]. It is based on intermediate waveguides with a cross-section evolving between the rectangular and the circular waveguides. In the present work, it consists of three sections. Section 1 is the nominal WR-28 standard rectangular waveguide at port 1, which allocates the dedicated port for the sum pattern (Σ). Section 2 is a waveguide with a bow-tie shape [55,56], which converts the $TE_{10}{}^{rec}$ mode of the input rectangular waveguide into the $TE_{11}{}^{cir}$ mode of the output circular waveguide. Section 3 includes stepped intermediate circular waveguides to reach the required final diameter of 10 mm ($1.27\;\lambda_{0}$) at port 2.

Although the design is narrow band (37.5–39 GHz) for this type of transition, the presented configuration is intended to have enough degrees of freedom to allow its integration with the mode converter for the difference pattern (Δ) described later. Moreover, the circular waveguide at the output is over-dimensioned to allow the propagation of the $TE_{01}{}^{cir}$ mode, which is a difference with respect to classical transitions between rectangular and circular waveguides.

The mode chart of the standard WR-28 rectangular waveguide is illustrated in Figure 2a. This waveguide supports a monomode propagation bandwidth between 21.07 GHz (cutoff frequency of the fundamental $TE_{10}{}^{rec}$ mode) and 42.14 GHz (cutoff frequency of the first degenerated $TE_{01}{}^{rec}\;\mathrm{and}\;TE_{20}{}^{rec}$ higher order modes). Figure 2b presents the mode chart of the circular waveguide of diameter 10 mm. The cutoff frequency of the fundamental $TE_{11}{}^{cir}$ mode is 17.58 GHz. In addition, this waveguide will have several potential propagating modes in the desired operating bandwidth (37.5–39 GHz). One of them is the $TE_{01}{}^{cir}$ higher order mode. The others are undesired modes whose levels will be either not excited by the symmetry of the structure, or with a level small enough to not degrade the system performance. The reason to work with the $TE_{11}{}^{cir}$ fundamental mode and at the same time with the $TE_{01}{}^{cir}$ mode, will be explained in detail in the next sections. Each mode will be associated to the sum (Σ) and difference (Δ) patterns, respectively.

It should be noted that the $TE_{mn}{}^{cir}$ and $TM_{mn}{}^{cir}$ modes with index m≠0 represent two independent modes, with $\sin\left(m\phi \right)\;\mathrm{and}\;\cos\left(m\phi \right)$ angular variation, respectively, in its longitudinal field component [57,58]. Thus, they have different E-field vector polarization. In the case of $TE_{11}{}^{cir}$, for example, one mode is aligned along the $\hat{x}$-axes ($TE_{11}{{}_{x}}^{cir}$), while the other is oriented along the $\hat{y}$-axes ($TE_{11}{{}_{y}}^{cir}$). As shown in Figure 3, the fundamental mode of the rectangular waveguide $TE_{10}{}^{rec}$ at port 1 with polarization along the $\hat{y}$ direction, will excite the fundamental mode of the circular waveguide $TE_{11}{{}_{y}}^{cir}$ at port 2, polarized along the same $\hat{y}$ direction.

The S-parameters for this mode converter are depicted in Figure 4. CST Studio Suite [59] was used for the simulations. $S_{11}\left(TE_{10}{}^{rec}\right)$ represents the return loss of the fundamental mode ($TE_{10}{}^{rec}$) within the standard rectangular waveguide. The result is better than 30 dB from 36 to 40 GHz, in a 10.5% bandwidth.

The structure avoids the generation of the modes without perfect magnetic wall boundary conditions at the symmetry plane [57,58]. Since symmetry can be degraded due to the tolerances in the fabrication, a full analysis is conducted as shown in Figure 4b. It includes all the modes within the band of interest according to Figure 2b, regardless of their symmetry, to be sure about their effect on the operating bandwidth. The cutoff frequency of the $TE_{21}{}^{cir}$ mode is around 49 GHz which is out of the operating frequency bandwidth. Thus, Figure 4b shows the generation of the first eight higher order modes in the circular waveguide, under the excitation of the mode converter by the $TE_{10}{}^{rec}$. It is clear that the higher order modes have been adequately attenuated, with the exception of the $TM_{11}{{}_{y}}^{cir}$. This mode exhibits a level worse than −30 dB, since the E-field is partially oriented in the same direction as the $TE_{11}{{}_{y}}^{cir}$. The peak observed around 36.5 GHz is precisely related to the cutoff frequency of $TM_{11}{{}_{y}}^{cir}$ mode. Any possible asymmetry in the structure could unintentionally increment this level.

### 2.2. Mode Converter from the $TE_{10}{}^{rec}$ Mode in the Rectangular Waveguide to the $TE_{01}{}^{cir}$ Mode in the Circular Waveguide

The proposed $TE_{10}{}^{rec}\;–\;TE_{01}{}^{cir}$ mode converter is shown in Figure 5. This topology, as presented in Figure 5a, follows the design principles described in [60]. It has two parts. The first part is a four-way power divider network, starting at the standard WR-28 waveguide. It allocates the dedicated port for the difference pattern (port 1, Δ). The second part is the junction combining the four signals coming from the first part, exciting the $TE_{01}{}^{cir}$ circular waveguide mode at the output (port 2). The operation is depicted in Figure 5b, with the electric field pattern for the modes involved in the mode conversion.

The four-way power divider is made up of three sections. Section 1 is the standard WR-28 rectangular waveguide for port 1, allocating the fundamental $TE_{10}{}^{rec}$ mode associated to the system difference pattern (Δ). Section 2 is a power divider, including stepped rectangular waveguides and a septum to divide the power evenly toward two outputs. Section 3 is a double power divider, again including an adapter and a septum, to achieve the four ways. Due to the symmetry of the four-way power divider network, all four outputs have the same phases and amplitudes. This leads to an azimuthal E-field pattern that easily couples to the $TE_{01}{}^{cir}$ mode in port 2. It should also be noticed that the locations and symmetries of the outputs in the four power dividers are crucial for preventing the generation of higher order modes.

The return loss for the ${TE}_{10}{}^{rec}$ ($S_{11}\;({TE}_{10}{}^{rec})$) in the rectangular waveguide is shown in Figure 6a. Its performance, as expected, is worse than for the mode converter in Figure 4, since it involves a more complex structure. Nevertheless, the return loss is better than 20 dB in the operating bandwidth from 37.5 to 39 GHz. The generation of higher order modes at the circular waveguide when the mode converter is excited by the ${TE}_{10}{}^{rec}$ is investigated in Figure 6b. It shows that all higher order modes, other than the $TE_{01}{}^{cir}$, are deeply attenuated when the $TE_{10}{}^{rec}$ fundamental mode of the rectangular waveguide is injected into the network.

## 3. Integration of the Dual-Mode Converter and the Conical Horn Antenna

This section presents the integration of the previous mode converters into the Dual-Mode Converter (DMC). Furthermore, the DMC will be also integrated later with a Conical Horn Antenna (CHA), leading to the Dual-Mode Conical Horn Antenna (DM-CHA).

### 3.1. Integration of the $TE_{10}{}^{rec}–TE_{11}{}^{cir}$ and $TE_{10}{}^{rec}–TE_{01}{}^{cir}$ Mode Converters for Obtaining the Dual-Mode Converter (DMC)

The previous Section 2 has introduced two mode converters operating separately. Now, both converters are combined and connected to each other to form a Dual-Mode Converter (DMC) as presented in Figure 7. The DMC is a device with three physical ports: two rectangular input ports working with their corresponding $TE_{10}{}^{rec}$ mode (nominal WR-28 rectangular waveguides in port 1 to excite the sum Σ pattern, and port 2 for the difference Δ pattern), and one circular output port of diameter 10 mm (port 3). This last port has two working modes, the $TE_{11}{}^{cir}$ and $TE_{01}{}^{cir}$, with cutoff frequencies of 17.5 GHz and 36.5 GHz, respectively (see Figure 2). Excitations by port 1 and port 2 generate the $TE_{11}{}^{cir}$ and $TE_{01}{}^{cir}$ modes, respectively.

As mentioned earlier, due to the possible asymmetries in the construction of the DMC, it is crucial that the two mode converter modules are precisely connected in order to prevent additional high-order mode excitation. Thus, the accurate placement of the four rectangular arms that are attached to the output is very important, in order to maximize the isolation between the two mode converters. It should be noted that the $TM_{11}{}^{cir}$ mode at the circular waveguide has the same cutoff frequency of 36.5 GHz as the $TE_{01}{}^{cir}$. Careful attention is required for this mode, which is sufficiently attenuated as seen in Figure 4b. At the upper end of the design band, the $TE_{21}{}^{cir}$ mode starts to propagate at 40 GHz. Thus, the dimensioning and design of the DMC allows its proper operation for the design band (37.5–39 GHz). The results for the DMC are presented later in Section 3.3.

### 3.2. Integration of the Dual-Mode Converter (DMC) with the Conical Horn Antenna (CHA) for Obtaining the Dual-Mode Conical Horn Antenna (DM-CHA)

The Dual-Mode Converter (DMC) allows the generation of the $TE_{11}{}^{cir}$ and $TE_{01}{}^{cir}$ modes from independent rectangular ports. In this section, the purpose is to integrate the DMC with a conical horn antenna (CHA), designed with an aperture diameter of $D_{horn}$ = 38.2 mm and a flare length of $L_{horn}$ = 64.4 mm. It is designed for the desired operating bandwidth, providing the required sum or difference pattern under the adequate excitation, as seen in Figure 8. An important parameter to be considered in the design is the length connecting the circular waveguide at the input of the CHA with the circular waveguide at the output of the DMC. The obtained length of L=32 mm, is important for controlling the mutual coupling between modes, and for the final return losses. This step finishes the design of the Dual-Mode Conical Horn Antenna (DM-CHA). It radiates the $TE_{11}{}^{cir}$ and $TE_{01}{}^{cir}$ modes which are excited by the DMC, to generate two independent diagrams: the sum (Σ) and difference (Δ) radiation patterns, respectively. They are shown in Figure 8.

### 3.3. Full-Wave Simulations of the DMC and the DM-CHA

The S-parameters of different simulations are represented in Figure 9. The six lines represent the reflection coefficient for the $TE_{10}{}^{rec}$ mode in the rectangular waveguide, but for different structures and excitations. The first and second lines in the figure legend are associated to the reflection coefficient of the two mode converters ($TE_{10}{}^{rec}\;–\;TE_{11}{}^{cir}$ and $TE_{10}{}^{rec}\;–\;TE_{01}{}^{cir}$) working independently, as in Figure 4 and Figure 6. The third and fourth lines in the legend are the reflection coefficients of the DMC, when it is excited by either port 1,Σ or by port 2,Δ, respectively. Finally, the fourth and fifth lines in the legend are the reflection coefficients of the DM-CHA (the DMC connected to the conical horn), when it is excited by either port 1,Σ or by port 2,Δ, respectively.

The return loss for the DMC and the mode converters working independently show good values. Nevertheless, some deterioration is observed in the DMC, especially in the difference pattern response (Δ), due to the mutual interaction of the two mode converters when they are combined. The return loss for the DM-CHA system in Figure 9 is better than 20 dB for excitation by $port\;1\;\left(\Sigma \right)$, and better than 9 dB for excitation by port 2 (Δ), across the whole frequency band from 37.5 to 39 GHz. This is a 0.5 GHz reduction compared to the desired 37 to 39 GHz band.

## 4. Experimental Results

Once the simulations had demonstrated the feasibility of the presented structure, the proposed antenna system was manufactured for its validation, and its performance was examined. The fabricated components of the DM-CHA system, including the DMC and the isolated CHA, are shown in Figure 10. The port names follow the same criteria used in Figure 7 and Figure 8. Subtractive manufacturing by Computer Numerical Control (CNC) was used to fabricate the DMC. This minimized possible fabrication asymmetries that would result in high-order mode excitation. Additive Manufacturing (AM) by Direct Metal Laser Sintering (DMLS) was used to fabricate the CHA, with a post-processing treatment of sandblasting to reduce the roughness of the metal powder.

The DMC network is made up of two parts. The lower part contains two standard WR-28 rectangular waveguide flanges (port 1 for the sum Σ pattern and port 2 for the difference Δ pattern) and the mode converters. The upper part serves as a cap to cover the feeding network. It also includes a small part of port 2 to complete it.

The measurement setup of the integrated DM-CHA system, which is connected to a two-port Vector Network Analyzer (VNA – Anritsu model MS46122B-040), is presented in Figure 11a. The S-parameters were measured and a comparative study was carried out between the simulated and measured results in Figure 11b. The return losses for the $S_{11}$ (sum Σ pattern) and $S_{22}$ (difference Δ pattern) are better than 20 dB and 9 dB, respectively, in the frequency range of 37.5 to 39 GHz. This shows a good agreement with the simulated results. Likewise, the measured isolations $S_{12}$, $S_{21}$ are better than 35 dB across the whole bandwidth, providing a satisfactory level of isolation between both the sum and difference ports. It should be noted that the simulated $S_{12}$, $S_{21}$, with perfect symmetries, are below −70 dB.

In the next step, the measurement setup in the anechoic chamber of Escuela Politécnica Superior of Universidad Autónoma de Madrid [61] is presented in Figure 12 to analyze the radiation patterns. Both sum (Σ) and difference (Δ) radiation patterns were generated by exciting subsequently their associated nominal WR-28 standard rectangular waveguide ports in the frequency bandwidth of 37.5–39 GHz. A waveguide matched load was connected to the port associated to the non-measured pattern.

The purity of the two modes generated by the DMC and their interaction with the CHA were checked, comparing them with the simulations for the radiation of the DM-CHA in Figure 8. Figure 13 shows the simulation and measurement results of the normalized radiation pattern in azimuth plane ($\emptyset =0^{{}^{\circ}}$, according to the coordinate system presented in Figure 8), at the three selected frequencies of 37.5, 38, and 38.5 GHz. A vertically polarized radiation is achieved for both sum (Σ) and difference (Δ) patterns, coherent with the E-field polarization of the $TE_{11}{{}_{y}}^{cir}$ mode in Figure 3, and the $TE_{01}{}^{cir}$ mode in Figure 5. A very good agreement is observed between the experimental and the simulated performance. Nevertheless, there is a remarkable difference between the two normalized radiation patterns in the main beam broadside direction ($\theta =0^{{}^{\circ}}$, null depth of the monopulse pattern), which is below −35 dB, −42 dB and −35 dB, respectively, and an almost symmetrical response around the broadside direction. These are very important parameters for adequate monopulse operation.

It is also noteworthy that the angular coverage range of the monopulse pattern is approximately $\pm 11^{{}^{\circ}}$ (the gray box in Figure 13). This is an important parameter for 5G high data rate and radiolocation satellite communications in the millimeter-wave band. This coverage is directly related to the direction of arrival of the received user signal from Earth, which is calculated by comparing the received signal levels from the difference and sum antenna patterns (monopulse function). The limit of the monopulse coverage is defined in the angular direction in which the sum and difference pattern levels are equal. The system is not operative in angular directions where the difference pattern level is higher than the sum pattern [2]. Wider angular coverage of the monopulse system would require multibeam or scanning (mechanical or electrical) beam capability. In addition, the simulated and measured results of the normalized radiation pattern in polar plot in azimuth ($\emptyset =0^{{}^{\circ}}$) and elevation ($\emptyset =90^{{}^{\circ}}$) planes have been presented in Figure 14.

Once the monopulse performance of the antenna system has been demonstrated, the aperture efficiency is studied. This parameter is the ratio of the effective radiating area to the physical area of the aperture. As it is explained in [62] (Section 13.5), the theoretical maximum achievable aperture efficiency in a conical horn antenna is 84%, under $TE_{11}{{}_{y}}^{cir}$ excitation with ideal uniform phase at the antenna aperture. This is impractical since it would require an infinite horn length $L_{horn}$. In Figure 15, the simulated and measured antenna directivities of the sum (Σ) radiation pattern are shown, as well as the related aperture efficiency based on the horn antenna aperture area size. The simulated directivity is higher than 22 dBi throughout the frequency band, with a peak value of 22.4 dBi at 38.5 GHz.

In terms of aperture efficiency, the antenna presents more than 70% in simulation. This value perfectly matches with the expected value for a conical horn antenna with the horn diameter-to-length ratio actually implemented ($D_{horn}/L_{horn}=0.6$) [62] (Section 13.5). These simulation values indicate that the $TE_{11}{{}_{y}}^{cir}$ mode of the sum (Σ) pattern has been properly generated. The corresponding measured values are directivity over 21.6 dBi, with a maximum of 21.9 dBi also at 38.5 GHz, and antenna aperture efficiency higher than 65%. The measured 0.5 dB directivity and 5% aperture efficiency reductions compared with simulations can be translated in a maximum E-field phase deviation of 15° at the entire antenna aperture area, perfectly explained by fabrication tolerances.

The simulated and measured antenna realized gain are also included in Figure 15. The conductivity considered in the simulations for the alloy material coming from the DMLS manufacturing of the CHA was 25,000 S/m. This value generates in simulations the same gain-to-directivity drops as in the measurements (0.3 dB average in the frequency range). Therefore, it can be considered as the most suitable value to represent the ohmic and roughness losses of the alloy material for the additive manufacturing process. Accordingly, the overall measured realized gain varies from 21 dBi to 21.7 dBi (dotted black line) while the measured directivity is from 21.6 dBi to 21.8 dBi, which means that the antenna radiation efficiency (which contains system losses) varies from 85% (39 GHz) to 97.7% (37.9 GHz). When the radiation and aperture efficiencies are both considered (some authors name it as total aperture efficiency [62] (Sections 2.8 and 14.5), the measured total antenna efficiency is higher than 60% up to 38.5 GHz, and always better than 50%. This is shown in Figure 15.

Therefore, based on all the above explained information, the simulation and measurement results show an excellent match, demonstrating the suitability of the proposed novel antenna concept.

Table 1 presents a comparison between the results achieved by the proposed antenna and those obtained in other state-of-the-art works within the same $k_{a}$ frequency band (27–40 GHz). The introduction of a fully metallic antenna showcases commendable directivity, particularly when considering its compact size ($4.84\;\lambda_{0}\times 4.84\;\lambda_{0}\times 13.9\;\lambda_{0}$). Notably, exceptionally high radiation efficiency has been achieved through the implementation of a fully metallic structure.

## 5. Conclusions

A novel concept of a Dual-Mode Conical Horn Antenna (DM-CHA) has been introduced in this paper for monopulse systems. It is designed for 5G millimeter-wave applications operating at 37.5–39 GHz. The DM-CHA system has been achieved by integration of a Dual-Mode Converter (DMC) network to a conical horn antenna (CHA). The DMC network includes two mode converters modules (${TE}_{10}{}^{rec}–{TE}_{11}{}^{cir}$ and ${TE}_{10}{}^{rec}–{TE}_{01}{}^{cir}$). The CHA connected to the DMC is used to excite either the ${TE}_{11}{}^{cir}$ or the ${TE}_{01}{}^{cir}$ modes, which radiate the sum (Σ) and difference (Δ) radiation patterns, respectively, involved in the monopulse system operation.

The proposed DM-CHA has been evaluated experimentally. It has been manufactured using two techniques. The DMC has been fabricated by subtractive manufacturing using Computer Numerical Control (CNC) technology. The CHA has been fabricated by additive manufacturing using Direct Metal Laser Sintering (DMLS) technology. After manufacturing the whole system, experimental measurements have been obtained for the S-parameters and for the normalized sum (Σ) and difference (Δ) radiation patterns. The measured results have been compared with the simulated results, with a very good agreement. Finally, the measured realized gain and directivity of the sum (Σ) radiation patterns have been compared. About 0.3 dB average loss in the frequency range has been observed for the measured results. This can be explained by the associated roughness of the DMLS process of the inner wall of the CHA.

Monopulse antennas operating in the millimeter-wave frequency band play a crucial role in advanced radar systems, particularly for tracking and guiding high-speed, precision-targeted objects. Furthermore, monopulse antennas in this frequency band fulfill a critical role in the detection and tracking of elusive targets, a task that proves challenging for lower-frequency radar systems. This capability holds paramount importance in countering the advanced technologies employed by potential adversaries. Overall, monopulse antennas in the millimeter-wave frequency band are essential components in modern technology, enabling precise and reliable tracking and guidance in complex and dynamic scenarios. According to the full set of experimental results, the novel monopulse antenna concept proposed in this work has shown its potential for future 5G millimeter-wave applications requiring monopulse operation.

## Figures and Tables

**Figure 1 sensors-23-08157-f001:**
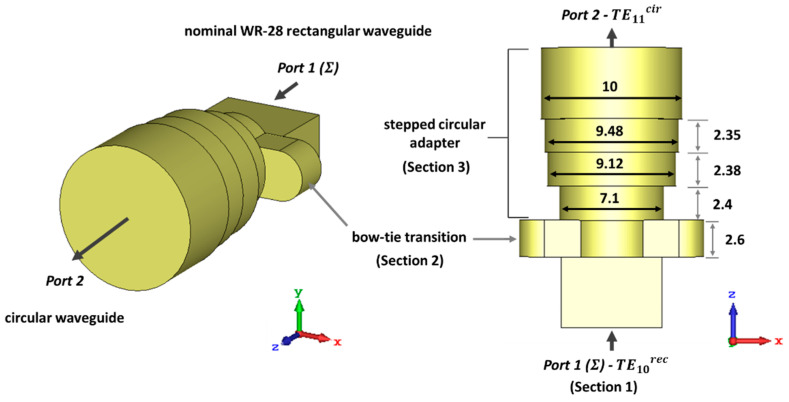
Schematic view of the mode converter from the $TE_{10}{}^{rec}$ mode in the rectangular waveguide (which will be later the sum port) to the $TE_{11}{}^{cir}$ mode in the circular waveguide (connected later to the conical horn).

**Figure 2 sensors-23-08157-f002:**
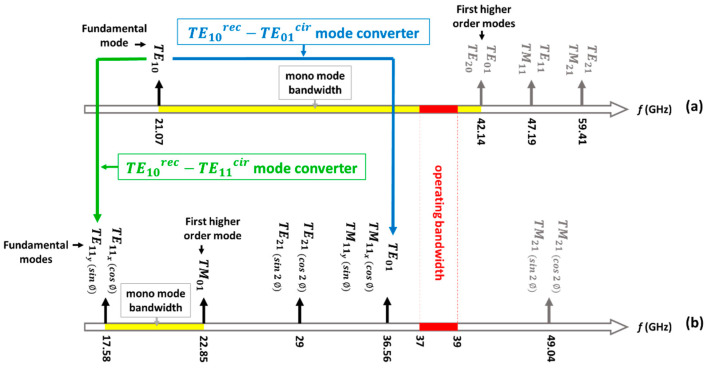
Cutoff frequency of the modes at the ports of the mode converters. (**a**) Nominal WR28 standard rectangular waveguide used for the dedicated ports associated to the sum (Σ) and difference (Δ) patterns. (**b**) Common circular waveguide used as input to the conical horn antenna (diameter of 10 mm).

**Figure 3 sensors-23-08157-f003:**
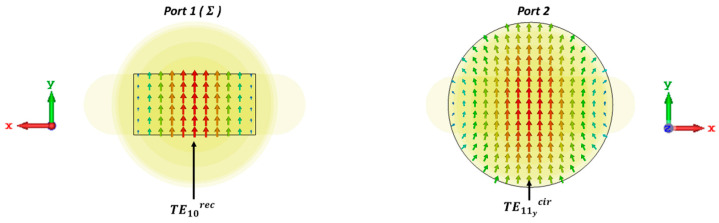
The electric field of the fundamental modes related to rectangular waveguide (${TE}_{10}{}^{rec},\;port\;1$) and circular waveguide (${TE}_{11}{{}_{y}}^{cir},\;port\;2$) with polarization along the $\hat{y}$ direction.

**Figure 4 sensors-23-08157-f004:**
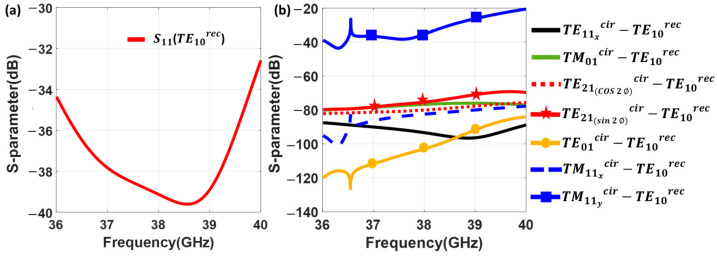
Simulated S-parameters of the mode converter in Figure 1, from ${TE}_{10}{}^{rec}$ mode in the rectangular waveguide to the ${TE}_{11}{}^{cir}$ mode in the circular waveguide. (**a**) Return loss at the rectangular waveguide for the ${TE}_{10}{}^{rec}$. (**b**) Generation of higher order modes at the circular waveguide when the mode converter is excited by the ${TE}_{10}{}^{rec}$ mode at the rectangular waveguide.

**Figure 5 sensors-23-08157-f005:**
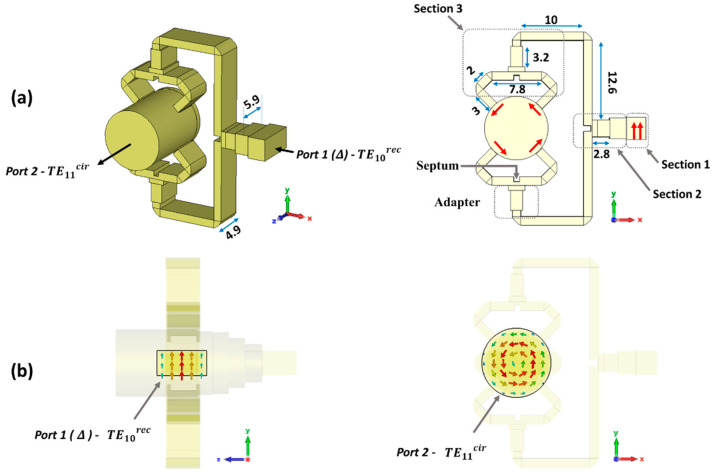
Schematic view of the mode converter from the ${TE}_{10}{}^{rec}$ mode in the rectangular waveguide to the ${TE}_{01}{}^{cir}$ mode in the circular waveguide. (**a**) Structure for the mode converter. (**b**) The electric field of the modes related to rectangular waveguide (port 1, Δ) and circular waveguide (port 2).

**Figure 6 sensors-23-08157-f006:**
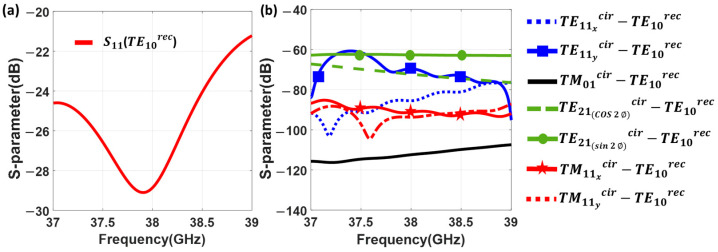
Simulated S-parameters of the mode converter in Figure 5, from $TE_{10}{}^{rec}$ mode in the rectangular to the $TE_{01}{}^{cir}$ mode in the circular waveguide. (**a**) Return loss at the rectangular waveguide for the $TE_{10}{}^{rec}$. (**b**) Generation of higher order modes at the circular waveguide when the mode converter is excited by the $TE_{10}{}^{rec}$ mode at the rectangular waveguide.

**Figure 7 sensors-23-08157-f007:**
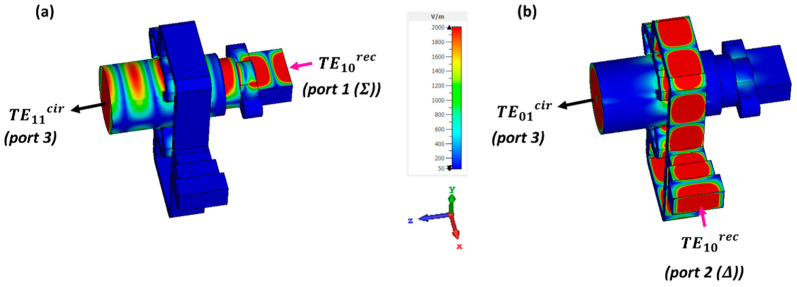
E-field simulation of the Dual-Mode Converter (DMC) network. (**a**) Excitation by the $TE_{10}{}^{rec}$ mode at the rectangular waveguide dedicated to the sum pattern (port 1,Σ), generating the $TE_{11y}{}^{cir}$ mode at the circular waveguide. (**b**) Excitation by the $TE_{10}{}^{rec}$ mode at the rectangular waveguide dedicated to the difference pattern (port 2,Δ), generating the $TE_{01}{}^{cir}$ mode at the circular waveguide.

**Figure 8 sensors-23-08157-f008:**
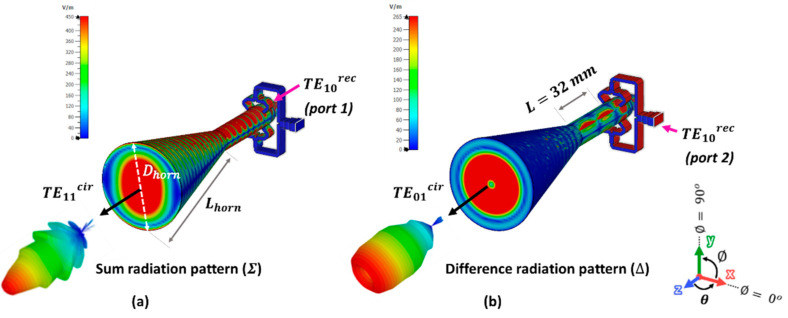
E-field simulation of the Dual-Mode Conical Horn Antenna (DM-CHA). (**a**) Excitation by the $TE_{10}{}^{rec}$ mode of port 1, generating the sum (Σ) radiation pattern. (**b**) Excitation by the $TE_{10}{}^{rec}$ mode of port 2, generating the difference (Δ) radiation pattern.

**Figure 9 sensors-23-08157-f009:**
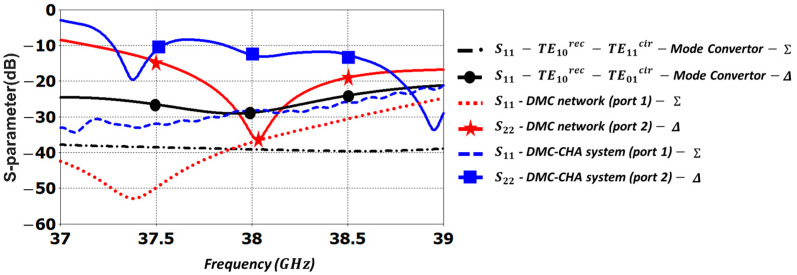
Simulated reflection coefficient for the ${TE}_{10}{}^{rec}$ mode: in the ${TE}_{10}{}^{rec}–{TE}_{11}{}^{cir}$ and ${TE}_{10}{}^{rec}–{TE}_{01}{}^{cir}$ mode converters working independently; in the Dual-Mode Converter (DMC); and in the Dual-Mode Conical Horn Antenna (DM-CHA).

**Figure 10 sensors-23-08157-f010:**
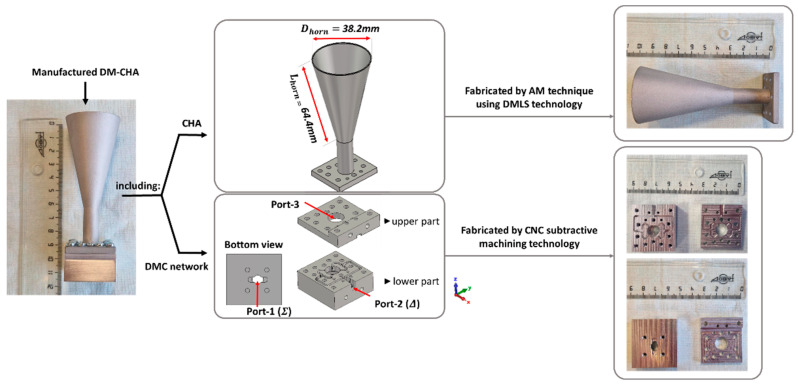
Manufacturing scheme of the Dual-Mode Conical Horn Antenna (DM-CHA). It includes the Dual-Mode Converter (DMC) and the Conical Horn Antenna (CHA).

**Figure 11 sensors-23-08157-f011:**
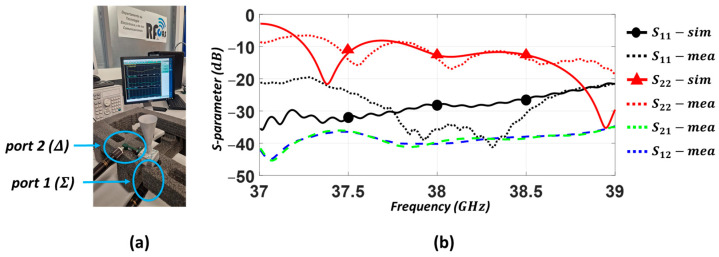
Dual-Mode Conical Horn Antenna (DM-CHA). (**a**) Measurement setup. (**b**) Measured and simulated S-parameters of the DM-CHA system.

**Figure 12 sensors-23-08157-f012:**
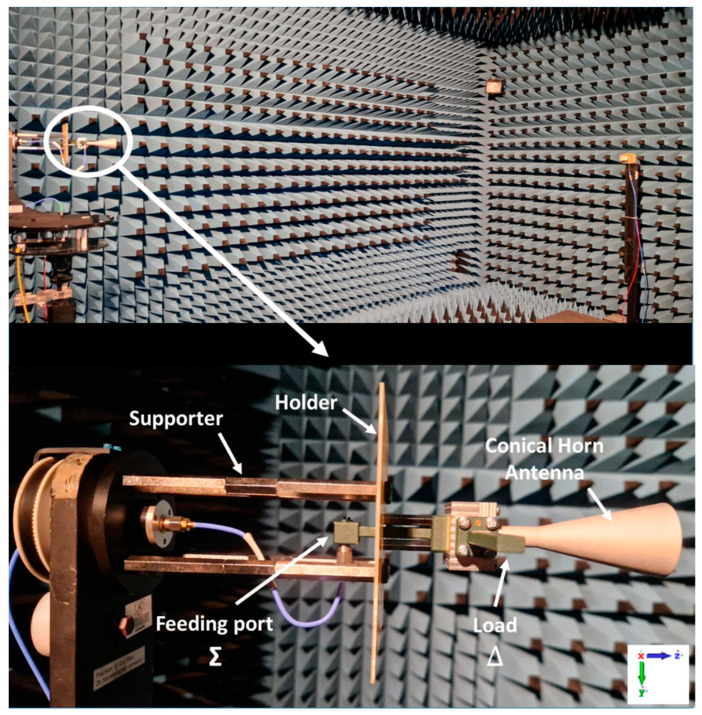
The measurement setup of the Dual-Mode Conical Horn Antenna (DM-CHA) in the anechoic chamber.

**Figure 13 sensors-23-08157-f013:**
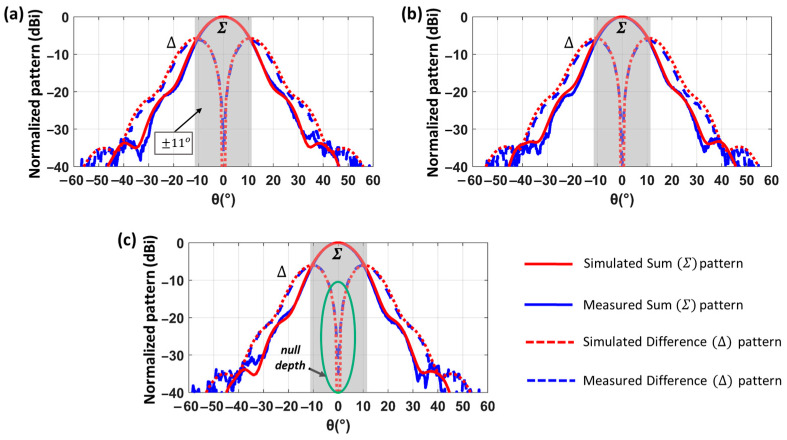
Simulation and measurement results of the normalized radiation pattern in azimuth plane ($\emptyset =0^{{}^{\circ}}$) at different frequencies: (**a**) 37.5 GHz, (**b**) 38 GHz and (**c**) 38.5 GHz.

**Figure 14 sensors-23-08157-f014:**
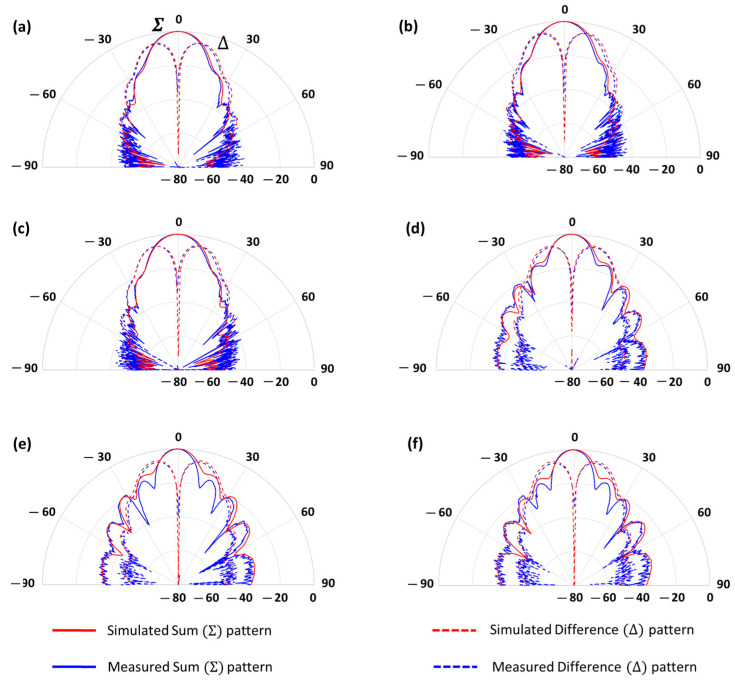
Simulation and measurement results of the normalized radiation pattern in polar plot in azimuth plane ($\emptyset =0^{{}^{\circ}}$) at different frequencies: (**a**) 37.5 GHz, (**b**) 38 GHz, (**c**) 38.5 GHz, and in elevation plane ($\emptyset =90^{{}^{\circ}}$) at different frequencies: (**d**) 37.5 GHz, (**e**) 38 GHz, (**f**) 38.5 GHz.

**Figure 15 sensors-23-08157-f015:**
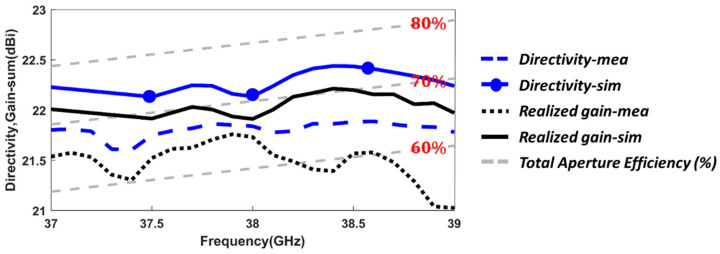
Simulation and measurement results of the Total Aperture Efficiency (%), realized gain, and directivity related to the Sum (Σ) radiation pattern.

**Table 1 sensors-23-08157-t001:** Comparison of the characteristics of this work with other state-of-the-art works.

Ref.	[35]	[28]	[19]	[20]	This Work
Technology	Horn + mode converter	Gap waveguide array	Metasurface	SIW slot array	Horn + mode converter
$f_{0}\;(\mathrm{GHz})$	35	30	33.5	31.25	38
Directivity [dBi]	24.08	27	31.2	18.74	21.8
Radiation efficiency (%)	NA	80	42	NA	85
Size $\left(\lambda_{0}{}^{3}\right)$	14.93×13.53×35.35	12×12×2.6	16.78×16.78	14.6×13.55	4.84×4.84×13.9
Full metal structure	Yes	Yes	No	No	Yes

*NA*: not applicable.

## Data Availability

No new data were created or analyzed in this study.

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
