# Peer review of "Dual-Mode Conical Horn Antenna with 2-D Azimuthal Monopulse Pattern for Millimeter-Wave Applications†"

_sensors, 2023, doi:10.3390/s23198157_

Round 1

Reviewer 1 Report

This study focuses on integrating a Dual Mode Converter (DMC) network with a high gain Conical Horn Antenna to enable monopulse operation in 5G millimeter-wave applications within the K-band (37.5 to 39 GHz). The system was designed to operate either the TE11 or TE 01 modes of a circular waveguide. The mode converters come from the TE10 mode of two separate WR-28 standard rectangular waveguide ports. A prototype was created using both subtractive manufacturing via Computer Numerical Control (CNC) technology and additive manufacturing through Direct Metal Laser Sintering (DMLS) technology. The system's performance was assessed through simulations and measurements, demonstrating strong alignment. This topic is relevant because of its implications for radar systems and possibly advancing 5G millimeter-wave technologies. 

The study doesn't present new design methods, but it shows a remarkable engineering effort in integrating different structures into a single system with some novelty. Although there are other works in the literature that present dual-mode systems with a coaxial horn, there is a relevant scientific contributing in this design.

Despite citing several references, the Introduction section does not clearly and objectively inform the reader of the state-of-the art. The Introduction alternates between trying to explain the model and citing other related works. This approach makes it difficult for the reader to understand the state-of-the art and, mainly, to understand the novelty of what is being presented.

The bibliographic review work can be improved regarding the state-of-the art of coaxial horns and dual-mode system integrated with horn antennas. Here are just some suggestions:

P. Zhang et. al., “Efficient design of axially corrugated coaxial-type multi-band horns for reflector antennas,” International Journal of Microwave and Wireless Technologies, vol. 9, no. 10, pp. 1975–1981, 2017.

E. Meyer et. al., "Miniaturized Conical Waveguide Filtenna for 5G Millimeter Wave Base Stations," 2021 15th European Conference on Antennas and Propagation (EuCAP), Dusseldorf, Germany, 2021, pp. 1-5, doi: 10.23919/EuCAP51087.2021.9411045.

E. Simionato et. al., "Novel coaxial transition for shaped coaxial horn antennas operating at millimeter wave frequencies," 2022 IEEE International Symposium on Antennas and Propagation and USNC-URSI Radio Science Meeting (AP-S/URSI), Denver, CO, USA, 2022, pp. 551-552, doi: 10.1109/AP-S/USNC-URSI47032.2022.9886866.

All figures should be improved, in particular fonts with a different size than the text font.

In several parts of the text and figures, the nomenclature S11, S22, etc. was used, or, worse, S1(1),1(1) - Despite being the format presented by the CST, the nomenclature is unclear. In fact, what it was presented is the module of the S-parameter (|S_nm|)

Minor editing of English language required.

Reviewer 2 Report

The authors demonstrated the Dual-Mode Conical Horn Antenna with 2-D Azimuthal Monopulse Pattern for Millimeter-Wave Applications. The concept is exciting, and the simulation results are reasonably good, showing potentially strong reconfigurability.

Overall, the article is good and well written, just enhance Figure 14 for better clarity.

There are minor errors that need to be checked!

Reviewer 3 Report

From line 69 onwards, it should be written in the next section, not in the introduction.

In Figure 1 and its dimensions, it is very effective in working frequency. Give an explanation compared to wavelength.

The ports are explained in an unintelligible manner. During the simulation, how it was simultaneously observed from the input of one mode and from the output of another mode.

In Figure 12, where is the farfield measurement probe?

Reviewer 4 Report

In this paper, authors presents novel concept of three-dimensional full metal system including a Dual Mode Converter (DMC) network integrated with a high gain Conical Horn Antenna (CHA).

The antenna is very conventional and these kinds of antennas have been widely studied in the literature. The  novelty and contribution of this work is very limited.

However, this is an article about antenna design, for this type of article it is written correctly and can be published after introducing a few changes.

Several elements need to be corrected in the paper:

(1)      The paper is well written and the presentation of the work is good.  However, the Figures quality is poor which needs improvement in the revised version.

(2)      Figure 4, Figure 6, Figure 9, Figure 11 and Figure 13 of poor quality.

(3)      In Figure 13, please present the results of the normalized radiation pattern in polar plot.

(4)      A lot of editing errors, e.g. S11. Please correct

(5)      The authors could compare their antenna with other antennas and refer to the obtained parameters.The authors must compare their work with other similar antennas operating at the same frequencies.

(6)      Radiation patterns: there is no definition of the reference system (coordinate system) in which the polarization of the antenna is determined.

(7)      Please present the radiation characteristics in the elevation plane.

The authors in the conclusions may add an exemplary practical application of the proposed antenna.

Please have the article read by a native speaker so that he or she can correct any language errors.

Round 2

Reviewer 3 Report

I accept this manuscript to be published as a paper.